# Sparse Reconstruction Using Hyperbolic Tangent as Smooth $l_1$-Norm Approximation

Hassaan Haider [1], Jawad Ali Shah [1,*], Kushsairy Kadir [2] and Najeeb Khan [3]

1   Department of Electrical and Computer Engineering, Faculty of Engineering and Technology,
    International Islamic University, Islamabad 44000, Pakistan
2   Electrical Section, British Malaysian Institute, Universiti Kuala Lumpur, Gombak 53100, Malaysia
3   Department of Electrical and Computer Engineering, University of British Columbia,
    Vancouver, BC V6T 1Z4, Canada
*   Correspondence: jawad.shah@iiu.edu.pk; Tel.: +92-333-5274448

**Abstract:** In the Compressed Sensing (CS) framework, the underdetermined system of linear equation (USLE) can have infinitely many possible solutions. However, we intend to find the sparsest possible solution, which is $l_0$-norm minimization. However, finding an $l_0$ norm solution out of infinitely many possible solutions is NP-hard problem that becomes non-convex optimization problem. It has been a practically proven fact that $l_0$ norm penalty can be adequately estimated by $l_1$ norm, which recasts a non-convex minimization problem to a convex problem. However, $l_1$ norm non-differentiable and gradient-based minimization algorithms are not applicable, due to this very reason there is a need to approximate $l_1$ norm by its smooth approximation. Iterative shrinkage algorithms provide an efficient method to numerically minimize $l_1$-regularized least square optimization problem. These algorithms are required to induce sparsity in their solutions to meet the CS recovery requirement. In this research article, we have developed a novel recovery method that uses hyperbolic tangent function to recover undersampled signal/images in CS framework. In our work, $l_1$ norm and soft thresholding are both approximated with the hyperbolic tangent functions. We have also proposed the criteria to tune optimization parameters to get optimal results. The error bounds for the proposed $l_1$ norm approximation are evaluated. To evaluate performance of our proposed method, we have utilized a dataset comprised of 1-D sparse signal, compressively sampled MR image and cardiac cine MRI. The MRI is an important imaging modality for assessing cardiac vascular function. It provides the ejection fraction and cardiac output of the heart. However, this advantage comes at the cost of a slow acquisition process. Hence, it is essential to speed up the acquisition process to take the full benefits of cardiac cine MRI. Numerical results based on performance metrics, such as Structural Similarity (SSIM), Peak Signal to Noise Ratio (PSNR) and Root Mean Square Error (RMSE) show that the proposed tangent hyperbolic based CS recovery offers a much better performance as compared to the traditional Iterative Soft Thresholding (IST) recovery methods.

**Keywords:** compressed sensing; holography cardiac cine MRI; $l_1$-norm smooth approximations; hyperbolic tangent function; soft thresholding



## 1. Introduction

Compressed Sensing (CS) exploits the sparsity of signals in a certain domain to find a near-optimal solution to the underdetermined system of linear equations. In CS, the sampling of signals depends on the information rate rather than its bandwidth. CS technique facilitates simultaneous acquisition and compression of compressible or sparse signals that potentially reduce the acquisition time. The CS is a data acquisition method that allows for the reconstruction of a signal from very few measurements if the signal is transformed in a sparsifying domain, and these measurements are highly incoherent with respect to its sparsifying transform. Unfortunately, most of the reconstruction techniques of compressively sampled signals are computationally expensive and non-linear [1–3].

CS has recently been used to reconstruct under-sampled biomedical images by exploiting the sparsity of biomedical images in the sparsifying domain. The Fourier-encoded nature of the MR image scanning process and the existence of suitable sparsifying transform domains, i.e., Wavelets, Contourlets, total variation, etc., make the MRI a potentially suitable application of CS [4]. Incoherent sampling, which is another important requirement of CS, can be accomplished with the variable density k-space sampling method to introduce noise-like random aliasing artefacts during the MR image recovery. Variable density k-space under-sampling pattern samples with high density from the center of the k-space that contains maximum energy of the MR images and undersamples the outer k-space region with lower density to efficiently reduce the MR image scanning time [5].

Cardiac cine magnetic resonance imaging (MRI) is an emerging medical imaging modality to evaluate the growth of Cardiac-Vascular Disease (CVD). It is useful in evaluating the cardiac wall thickness and motion in CVD patients [6–8]. Further, cardiac cine MRI aids in performing the quantitative study of ejection fraction and cardiac output of the heart. The ejection fraction is the percentage of blood that is ejected out of the ventricles with each contraction. This amount is used to determine heart failures and other types of heart diseases [9]. Cardiac output measures the amount of blood pumped by the heart per minute. However, these advantages are limited by the lengthy acquisition process of cardiac cine MRI that requires multiple breaths-holds of the patient and extended patient engagement in MRI scanner. Therefore, it is essential to accelerate the image acquisition in cardiac cine MRI by using fast pulse sequences and/or by reducing the number of samples taken during data acquisition [10,11]. As the former approach is inherently limited by different constraints, much research interest is moved to the latter approach. CS can be applied successfully to the cardiac cine MRI, where sparsity is exploited in the temporal dimension [4,12]. However, improving speed and efficiency of CS recovery methods is an active area of interest for researchers working in medical imaging especially MRI. The key conditions for the CS framework to work are sparsity, non-linear reconstruction and incoherent undersampling.

In MR imaging, sparsity can be accomplished by transforming the image in its sparse representation. To fulfil the condition of incoherent sampling in MR imaging, various undersampling patterns can be utilized, such as radial lines sampling and variable density sampling [4]. Non-linear reconstruction numerical techniques involve $l_1$-norm regularization in order to find sparse solutions to the least-squares optimization problem. $l_2$-norm based regularization provides the linear and simplest solution to under-determined system problem. However, it minimizes the energy of the error and distributes it over all solution set that results in non-sparse solution that does not fits in CS framework. Similarly $l_p$-norm $(1 < p < \infty)$ based regularization, as value of p starts growing it tends to penalize only the largest parameter, such as max function, and some bad parameters may hide under the largest parameters, which results in less-sparse solution. For this very reason $l_1$-norm is the preferred regularization, as it promotes sparsity that perfectly fits in CS framework [1–3]. However, $l_1$-norm penalty is non-differentiable, so applying efficient optimization methods that involve derivative are not feasible. Therefore, various methods have been proposed to resolve the $l_1$-norm regularization problem. The IST based recovery methods have successfully been utilized to efficiently reconstruct images from under-sampled data in the CS framework [5,13]. An iterative hard thresholding-based recovery method is proposed for the compressed sensing problem [14]. However, this algorithm has limited performance as compared to the soft thresholding-based methods. Random filters for compressive sampling [15], Bregman iterative algorithms for compressed sensing [16], and a weighted $l_1$ minimization recovery algorithm [17] are proposed to solve the compressed sensing recovery problem. Lately, smooth $l_1$-norm penalty based sparse signal reconstruction method was evolved for approximation of $l_1$-norm that uses a hyperbolic tangent function [18]. The research shows that this technique can be used for the reconstruction of undersampled MR images from fewer acquired samples, which allows fast imaging without compromising spatial resolution. Jawad et al. proposed that the wavelet thresholding can be implemented

using hyperbolic tangent function. It was explained that the differentiable hyperbolic function provides a much more accurate recovery than IST techniques [19–21]. It was experimentally shown that the hyperbolic tangent function performs a much improved signal recovery as compared to the hard threshold, soft threshold and Garrote threshold functions [22,23].

In this paper, CS technique is applied to reduce MRI scanning time, CS exploits the sparsity of MRI in Fourier domain that enables us to take fewer samples without compromising on the quality of image recovered from undersampled MRI. We propose a novel and more efficient CS recovery algorithm based on the hyperbolic tangent function for approximating $l_1$-norm and shrinkage operation for accurate recovery of compressively sampled sparse signals, MR images and the cardiac cine MRI. We introduce smooth approximation of $l_1$-norm, the hyperbolic tangent function, where steepest descent algorithm is applicable for minimization of objective function. The error bounds for the proposed $l_1$-norm penalty are presented in this paper. We have used the soft thresholding technique based on the hyperbolic tangent function that is inspired by the maximum a posteriori (MAP) noise estimator. In this work, we also recommend the efficient criteria for the tuning parameters. Performance analysis of the proposed method is shown using simulations; to recover random 1-D sparse signal, 2-D MR image and clinical cardiac cine MRI. Several quantitative performance measures are used apart from qualitative depiction, i.e., Mean Square Error (MSE), Root Mean Squared Error (RMSE), Signal to Noise Ratio (SNR), Peak Signal to Noise Ratio (PSNR), Improved Signal to Noise Ratio (ISNR), correlation, fitness, and Structural Similarity (SSIM) in order to prove the supremacy of proposed method over existing recovery techniques.

## 2. Materials and Methods

Reconstruction of undersampled signal through CS is an optimization problem, which promotes sparsity in our solution by minimizing the $l_1$–norm.

### 2.1. Proposed Method

Let $z \epsilon \mathbb{R}^n$ be the signal in a vector form and $y \epsilon \mathbb{C}^m$ be the undersampled measurements. Then, the CS recovery function is written as:

$$f(z) = \frac{1}{2}\|y - \Phi\Psi^H z\|_2^2 + \lambda\|z\|_1 \tag{1}$$

where $\Phi$ is the sampling domain of the signal $z$, whereas $\Psi$ represents sparsifying transform. The tuning parameter $\lambda$ in Equation (1) provides an important trade-off parameter between fidelity and sparsity. The performance of our algorithm is dependent on proper threshold level selection. We have employed the fixed value expression, depending on the signal dimensions and its noise variance [13].

$$\lambda = \sigma_v\sqrt{2\ln(n)} \tag{2}$$

where $\sigma_v$ is the noise standard variance and $n$ is the length of the sparse signal.

Since the hyperbolic tangent function has properties, such as non-convex, odd, smooth analytical bounded function that is monotonically increasing, the slope of the function at the origin can be tuned to any desired value [21]. So, our proposed approximation for the $l_1$ norm in Equation (1) is defined as:

$$\|z\|_1 \cong \sum_{i=1}^{n} z_i tanh(\gamma z_i) \tag{3}$$

Since the hyperbolic tangent function is used as a smooth and differentiable approximation to $l_1$-norm. Therefore, the value of $\gamma$ is taken quite high to make it closer to $l_1$-norm, as shown in Figure 1. It is also providing the benefits of the smoothness and differentiability. Equation (1) can now be written as:

$$f(z) = \frac{1}{2}\|\mathbf{y} - \Phi\Psi^{\mathrm{H}}\mathbf{z}\|_2^2 + \lambda \sum_{i=1}^{n} z_i tanh(\gamma z_i) \tag{4}$$

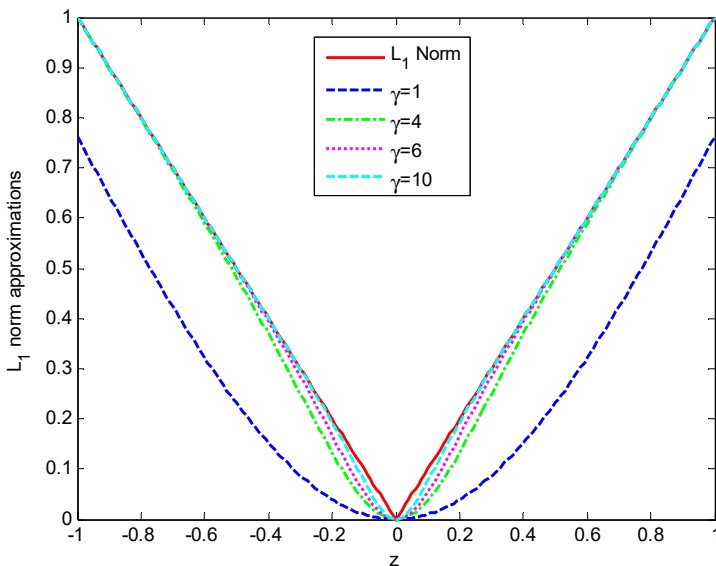

**Figure 1.** $l_1$ norm approximation using the hyperbolic tangent function for different values of $\gamma = (1, 4, 6,$ and $10)$. As the value of gamma continues to increase and the approximation is closer to the actual $l_1$ norm, however, it is less smooth. The proposed technique gives us the flexibility to choose between the level of smoothness and accuracy.

For steepest descent algorithm, vector differentiation can not be used. Hence, it can be rewritten as Equation (4) in element form to find the partial derivative. Let $\mathbf{A} = \Phi\Psi^{\mathrm{H}}$, then the element-wise equation is defined as:

$$f(z) = \frac{1}{2}\sum_{i}(\mathbf{Az} - y)_i(\mathbf{Az} - y)_i + \lambda z_i tanh(\gamma z_i) \tag{5}$$

Let $\mathbf{A} = \phi\Psi^{\mathrm{H}}$, then partial derivative of Equation (4) in element form is formulated as:

$$\frac{\partial f(z)}{\partial z_l} = \sum_{ij}\mathbf{A}_{ij}\mathbf{A}_{il}z_j - \sum_{i}y_i\mathbf{A}_{il} + \lambda\left(\tanh(\gamma z_l) + z_l\gamma\left(1 - tanh^2(\gamma z_l)\right)\right) \tag{6}$$

Hence, the steepest descent algorithm for $l$th update is:

$$(\Delta z)_l = -\eta\frac{\partial f(z)}{\partial z_l} \tag{7}$$

Equation (7) is used to find a solution using the steepest descent algorithm.

### 2.2. Error Bounds for Proposed Smooth $l_1$-Norm

The error bounds for the proposed smooth $l_1$ norm approximation defined by Equation (2) are derived in this section [24]. The $l_1$ norm approximation is proposed based on the following two principles.

1.  $|z| = (z)_+ + (-z)_+$, where $(z)_+ = \max\{z, 0\}$ is the plus function;
2.  This plus function can be smoothly approximated as:

$$(\mathbf{z})_+ \approx p(\mathbf{z}, \gamma) = \frac{1}{2}[\mathbf{z} + \mathbf{z}.\tanh(\gamma\mathbf{z})] \tag{8}$$

From Equation (8), we can write a smooth approximation of $l_1$ norm:

$$
\begin{aligned}
\|z\|_1 &= (z)_+ + (-z)_+ \approx p(z,\gamma) + p(-z,\gamma) \\
&= \tfrac{z}{2}[1 + \tanh(\gamma z)] - \tfrac{z}{2}[1 + \tanh(-\gamma z)] \\
&= \tfrac{z}{2}\tanh(\gamma z) + \tfrac{z}{2}\tanh(\gamma z) \\
&= z\,tanh(\gamma z) \\
&= \|z\|_\gamma
\end{aligned}
\tag{9}
$$

Equation (9) represents the $\gamma$ approximation of the $l_1$ norm, as shown in Figure 1.

Unlike the $l_1$ norm, we can apply the unconstraint optimization techniques, where gradient needs to be calculated and the proposed approximation is twice differentiable and the 1st and 2nd order gradients of the proposed $l_1$ norm are shown in Equations (10) and (11), respectively.

$$
\nabla(\|z\|) \approx \tanh(\gamma z) - \gamma z \left( tanh(\gamma z)^2 - 1 \right)
\tag{10}
$$

$$
\nabla^2(\|z\|) \approx 2\gamma(\gamma z\, tanh(\gamma z) - 1)\left( tanh(\gamma z)^2 - 1 \right)
\tag{11}
$$

As the value of $\gamma$ approaches infinity the error between $\|z\|_1$ and $\|z\|_\gamma$ approaches zero. We here propose the simple lemma to determine the error bounds for $\|z\|$ and $\|z\|_\gamma$.

**Lemma 1.** *The proposed smooth function of $l_1$ norm $f(z) = z\,tanh(\gamma z)$ fulfils the sufficient and necessary convexity condition in the interval $z \in [-1,1]$ as its derivative $f\prime(z)$ defined by Equation (9) is monotonically non-decreasing and its second derivative $f''(z)$ defined by Equation (10) is nonnegative for $0 < \gamma \leq 1$;*

**Lemma 2.** *$l_1$ norm approximation error bounds for any $z \in \mathbb{R}$ and $\gamma > 0$.*

$$
\left| \|z\|_1 - \|z\|_\gamma \right| \leq \frac{1}{2\gamma}
\tag{12}
$$

**Proof.** Let us consider two cases, first case for $z > 0$,

$$
\begin{aligned}
p(z,\gamma) - (z)_+ &= \tfrac{z}{2}(1 + \tanh(\gamma z)) - z \\
&= \tfrac{z}{2}(1 + \tanh(\gamma z)) - z \\
&= \tfrac{z}{2}(\tanh(\gamma z) - 1)
\end{aligned}
\tag{13}
$$

Now, we can find the maximum value of $\tanh(\gamma z)$ to find the upper bound for Equation (13). As we know that the maximum value of $\tanh(\gamma x)$ is 1, so we can write:

$$
\underset{z}{maxima}\ tanh(\gamma z) = \frac{e^{\gamma z} - e^{-\gamma z}}{e^{\gamma z} + e^{-\gamma z}} = 1
\tag{14}
$$

Using Equation (14), the relationship between $\gamma$ and $z$ can be easily derived as:

$$
z = \frac{1}{2\gamma}
\tag{15}
$$

By inserting the value of $z$ from Equation (15) in Equation (14)

$$
p(z,\gamma) - (z)_+ \leq \frac{1}{4\gamma}
\tag{16}
$$

$\square$

Figure 2 shows the proposed method obeys error bounds define by Equation (16).

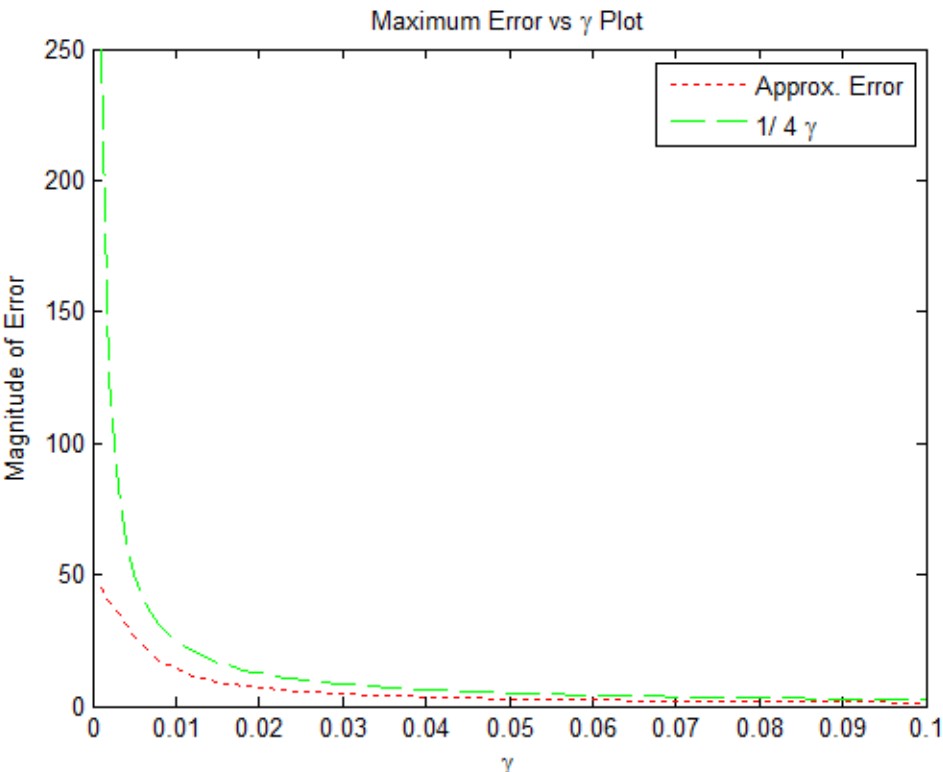

**Figure 2.** $l_1$ norm approximation error bounds for $z > 0$, the green line shows the upper bound proved mathematically in Equation (16), whereas the dotted red line shows the actual error between the proposed $l_1$ norm smooth approximation and actual non-differentiable $l_1$ norm. The error is maximum at approximately zero and approaches zero as $\gamma \to \infty$.

Figure 3 shows graphically that our proposed approximation function obeys the bounds defined by Equation (19).

For $z \leq 0$,

$$
\begin{aligned}
0 \leq p(z,\gamma) - (z)_+ \quad &= p(z,\gamma) \leq p(0,\gamma) \\
&= \tfrac{z}{2}\left(\tanh(\gamma z) - 1\right) \leq 0 \\
&= \tfrac{1}{4\gamma}
\end{aligned}
\tag{17}
$$

As $p$ is the monotonically increasing function. Hence, from Equations (17) and (18), $p(z,\gamma)$ will dominate $(z)_+$, so

$$
|p(z,\gamma) - (z)_+| \leq \frac{1}{4\gamma}
\tag{18}
$$

From Equation (8), we can insert $\|z\| = (z)_+ + (-z)_+$

$$
\begin{aligned}
\left|\|z\|_1 - \|z\|_\gamma\right| &= \left|p(z,\gamma) + p(-z,\gamma) - \left((z)_+ + (-z)_+\right)\right| \\
&\leq \left|p(z,\gamma) - (z)_+\right| + \left|p(-z,\gamma) - (-z)_+\right| \\
&\leq \tfrac{1}{4\gamma} + \tfrac{1}{4\gamma} = \tfrac{1}{2\gamma}
\end{aligned}
\tag{19}
$$

Figure 3 shows the error bounds versus error in the smooth approximation.

Let us define $\|z\|_{(1,\gamma)}$ as a smooth approximation to the $l_1$ norm function $\|z\|_1$ for a vector $z \in \mathbb{R}^n$ as:

$$
\begin{aligned}
\|z\|_{(1,\gamma)} &= \Sigma_i^n \|z_i\|_\gamma \\
\left|\|z\|_{(1,\gamma)} - \|z\|_1\right| &\leq 2n\tfrac{1}{4\gamma} = \tfrac{n}{2\gamma}
\end{aligned}
\tag{20}
$$

Hence, we can conclude that:

$$\lim_{\gamma \to \infty} \|z\|_{(1,\gamma)} = \|z\|_1 \ \forall z \in \mathbb{R}^n \tag{21}$$

Let $L : \mathbb{R}^n \to \mathbb{R}$ by any continuous cost function and defined by $f(z) = L(z) + \|z\|_1$ and $f_\gamma(z) = L(z) + \|z\|_{(1,\gamma)}$. If we define $\bar{z} = \overset{argmin}{z} f(z)$ and $\bar{z}_\gamma = \overset{argmin}{z} f_\gamma(z)$. By definition of $f$ and $f_\gamma$ and from Equation (20), it can be concluded that

$$\lim_{\gamma \to \infty} f_\gamma(z) = f(z) \ \forall z \in \mathbb{R}^n \tag{22}$$

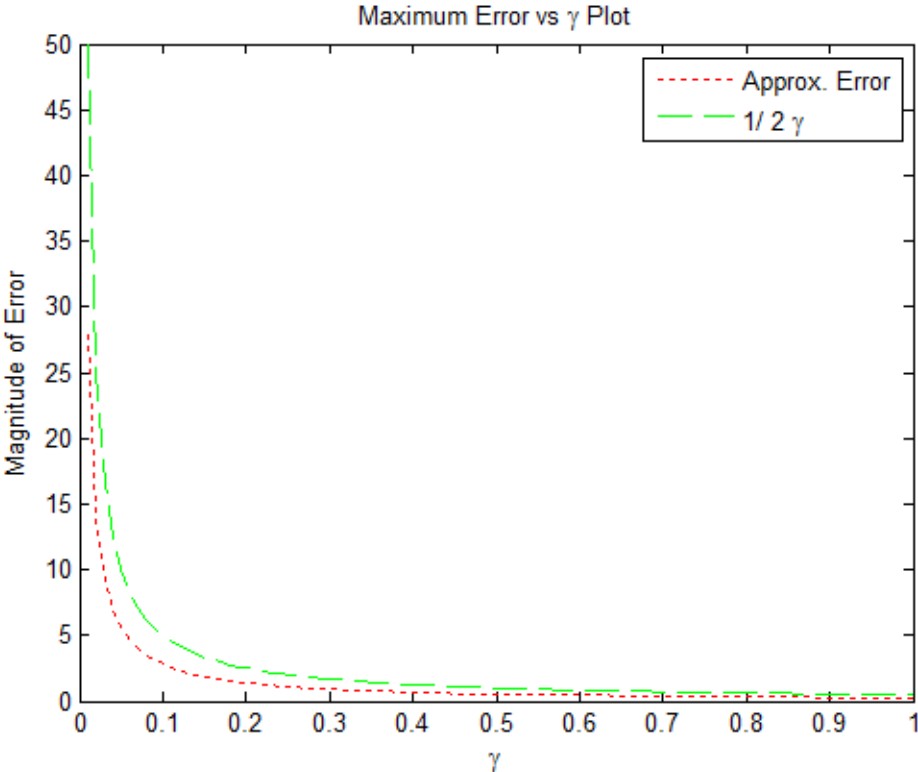

**Figure 3.** $l_1$ norm approximation error bounds for $z < 0$, the green line shows the upper bound proved mathematically in Equation (19), whereas the dotted red line shows the actual error between proposed $l_1$ norm smooth approximation and actual non-differentiable $l_1$ norm. The error is maximum at approximately zero and approaches zero as $\gamma \to \infty$.

In addition, it is a known fact that $f(\bar{z}) \leq f(z) \forall z$. In particular $f(\bar{z}) \leq f(\bar{z}_\gamma)$, then:

$$
\begin{aligned}
f(\bar{z}) \leq f(\bar{z}_\gamma) \ &= L(\bar{z}_\gamma) + \|\bar{z}_\gamma\|_1 \\
&= L(\bar{z}_\gamma) + \|\bar{z}_\gamma\|_1 + \|\bar{z}_\gamma\|_{(1,\gamma)} - \|\bar{z}_\gamma\|_{(1,\gamma)} \\
&= \left( L(\bar{z}_\gamma) + \|\bar{z}_\gamma\|_{(1,\gamma)} \right) + \left( \|\bar{z}_\gamma\|_1 - \|\bar{z}_\gamma\|_{(1,\gamma)} \right) \\
&= f_\gamma(\bar{z}_\gamma) + \left( \|\bar{z}_\gamma\|_1 - \|\bar{z}_\gamma\|_{(1,\gamma)} \right)
\end{aligned}
\tag{23}
$$

This implies that $f(\bar{z}) - f_\gamma(\bar{z}_\gamma) \geq -\frac{n}{2\gamma}$ from Equation (21), similarly $(\bar{z}) - f_\gamma(\bar{z}_\gamma) \leq \frac{n}{2\gamma}$, hence proved that $\lim_{\gamma \to \infty} f_\gamma(\bar{z}) = f(\bar{z})$.

It can be further stated that:

$$
\begin{aligned}
|f(\bar{z}_\gamma) - f(\bar{z})| \ &= |f(\bar{z}_\gamma) - f(\bar{z}) - f_\gamma(\bar{z}_\gamma) + f_\gamma(\bar{z}_\gamma)| \\
&\leq |f(\bar{z}_\gamma) - f_\gamma(\bar{z}_\gamma)| + |f_\gamma(\bar{z}_\gamma) - f(\bar{z})|
\end{aligned}
\tag{24}
$$

Hence, it proved that $\lim_{\gamma \to \infty} f(\bar{z}_\gamma) = f(\bar{z})$. Moreover, if $L$ is strictly convex, it can be easily proven that: $\lim_{\gamma \to \infty} z_\gamma = \bar{z}$.

### 3. The Map Estimator and Proposed Thresholding Mechanism

Conventionally $l_1$-norm minimization has inherent soft thresholding [13]. However, when we approximate the $l_1$-norm by the hyperbolic tangent function, thresholding is not done implicitly. The hard thresholding operator proposed in [14] can be defined by the following equation.

$$S_\beta(z) = \begin{cases} z & |z| > \beta \\ 0 & otherwise \end{cases} \tag{25}$$

We have used a new thresholding function based on the tangent hyperbolic function. Therefore, $\beta$ is an important parameter in controlling the under-sampling noise, which has Gaussian distribution [21]. To obtain the optimum value of $\beta$, the thresholding depends upon the undersampling noise. Therefore, the following data-driven thresholding parameter $\beta$ is used [13,25].

$$\beta = \frac{\sigma_v^2}{\sigma_z} \tag{26}$$

With $\sigma_z$ is the standard deviation of sparse signal and $\sigma_v$ the standard deviation of Gaussian-like noise produced due to under-sampling.

To enhance the performance under different scenarios, different mathematical thresholding operators could be found in the literature [26–28]. The main idea in this approach is mapping the values nearer to the origin to zero and those that are further away from the origin are shrunk towards zero.

The basic denoising technique aims to find the estimate of original image or signal from its perturbed set of observations, as shown in Equation (27).

$$y = z + v \tag{27}$$

where $y \in \mathbb{R}^n$ is the noisy image, $z \in \mathbb{R}^n$ is the original signal and $v$ is the zero-mean Gaussian noise with probability distribution function (pdf) given by:

$$p_v(\theta) = \frac{1}{\sqrt{2\pi\sigma_v^2}} \exp\left( \frac{\|\theta\|_2^2}{2\sigma_v^2} \right) \tag{28}$$

By taking the Wavelet transform of Equation (27), we get:

$$q = s + v \tag{29}$$

where $q = \Psi y$ and $s = \Psi z$ represent the sparsifying domain for noisy image and the original image, respectively. As Wavelet transform is the linear operator, therefore the zero-mean Gaussian noise $v$ after transformation will not change. The MAP estimation of random vector $s$ is given by:

$$\hat{s} = \max_{s \in \mathbb{R}^n} p(s|q) \tag{30}$$

By using Bayes' rule, one can ignore $p(q)$ as it is independent of $s$, MAP estimator can be written as:

$$\hat{s} = \max_{s \in \mathbb{R}^n} p(q|s) p_s(s) \tag{31}$$

The problem defined in Equation (31) can be further simplified by taking $p(q|s) = p_v(q - s)$ :

$$\begin{aligned} \hat{s} &= \max_s [p_v(q - s)] p_s(s) \\ &= \max_s [ln p_v(q - s) + ln p_s(s)] \\ &= \max_s \left[ ln \left\{ \frac{1}{\sqrt{2\pi\sigma_v^2}} \exp\left( -\frac{\|q-s\|_2^2}{2\sigma_v^2} \right) \right\}_n + ln p_s(s) \right] \\ &= \max_s \left[ ln \left\{ \left( \frac{1}{\sqrt{2\pi\sigma_v^2}} \right)^n \exp\left( -\frac{\|q-s\|_2^2}{2\sigma_v^2} \right) \right\} + ln p_s(s) \right] \\ &= \max_s \left[ -\frac{\|q-s\|_2^2}{2\sigma_v^2} + f(s) \right] \end{aligned} \tag{32}$$

where $f(s) = \ln p_s(s)$. By differentiating the argument of Equation (32) w.r.t. $s$ and equating the result to zero, we can calculate the MAP estimator for Wavelet coefficients of the noise-free image as:

$$\frac{(q_i - \hat{s}_i)}{\sigma_v^2} + f'(\hat{s}_i) = 0, \quad 1 \leq i \leq n \tag{33}$$

The pdf of biomedical images are more peaked at the center than Gaussian, so Laplacian can better estimate the distribution of Wavelet domain coefficients, i.e.,

$$p_s(s_i) = \frac{1}{\sqrt{2}\sigma_v} \exp\left(\frac{\sqrt{2}}{\sigma_v}|s_i|\right) \tag{34}$$

gives $f'(\hat{s}_i) = -\frac{\sqrt{2}}{\sigma} sig(\hat{s}_i)$. Solving Equation (33) will result in

$$q_i = \hat{s}_i + \sqrt{2}\, sig(\hat{s}_i) \tag{35}$$

Let $\beta = \sqrt{2}\sigma_v^2$ and solve Equation (35) for $\hat{s}_i$ to formulate the nonlinear shrinkage:

$$\hat{s}_i = S_\beta(q) = max\{|q| - \beta, 0\}.sig(q) \tag{36}$$

Equation (36) can be further elaborated as:

$$S_\beta(q) \cong \begin{cases} sgn(q)(|q| - \beta) & |q| > \beta \\ 0 & otherwise \end{cases} \tag{37}$$

In this paper, novel thresholding approach has been proposed, which is used on the hyperbolic tangent, as the hyperbolic tangent function slope can be adjusted from the origin and it is a bounded function that makes it an suitable surrogate function for soft thresholding. Hence, hyperbolic tangent based soft thresholding can be described mathematically by following equation:

$$S_\beta(q) \cong \begin{cases} cz\{tanh(\alpha(|q| - \beta))\} & |q| > \beta \\ 0 & otherwise \end{cases} \tag{38}$$

where $\beta$ is a thresholding parameter and parameter $\alpha$ is used to control the shape of the hyperbolic tangent function. If $\alpha$ is closer to zero, Equation (38) approximately changes into the soft thresholding function. When $\alpha$ approaches $\infty$, Equation (38) changes to the hard thresholding function, as shown in Figure 4. Our proposed Algorithm 1 starts as a soft thresholding function and smoothly changes to a hard threshold at higher iterations. The proposed approximation of this soft thresholding results in a better reconstruction as compared to the conventional soft thresholding method [21] as illustrated in Figure 4.

---

**Algorithm 1.** Proposed Algorithms.

---

**Inputs:**
Sensing matrix $\mathcal{F}_u$, measurement vector $y \epsilon \mathbb{C}^m$, parameters $\gamma$, $\lambda$ and $\beta$,
**Output:**
A k-sparse vector $\hat{\mathbf{x}} \in R^n$
Initialization: Initialize $\mathbf{x}_0$, Index $i = 0$
Step-1 (**Sparse Representation):** $z_i = \mathbf{\Psi} x_i$
Step-1 (**Gradient Computation**): Find $\nabla f(z_i)$ using Equation (5)
Step-2 (**Solution Update**): Compute the update using Equations (6) and (7).
Step-3 (**Shrinkage**): Estimate Solution using Equation (38), i.e., $\hat{\mathbf{z}}_{i+1} = S_\beta(\mathbf{z}_{i+1})$
Step-4 (**Repeat**): If stopping criterion is not met, $i = i + 1$ & go to step 1
Output: $\hat{x} = \mathbf{\Psi}^H \hat{z}_i$

---

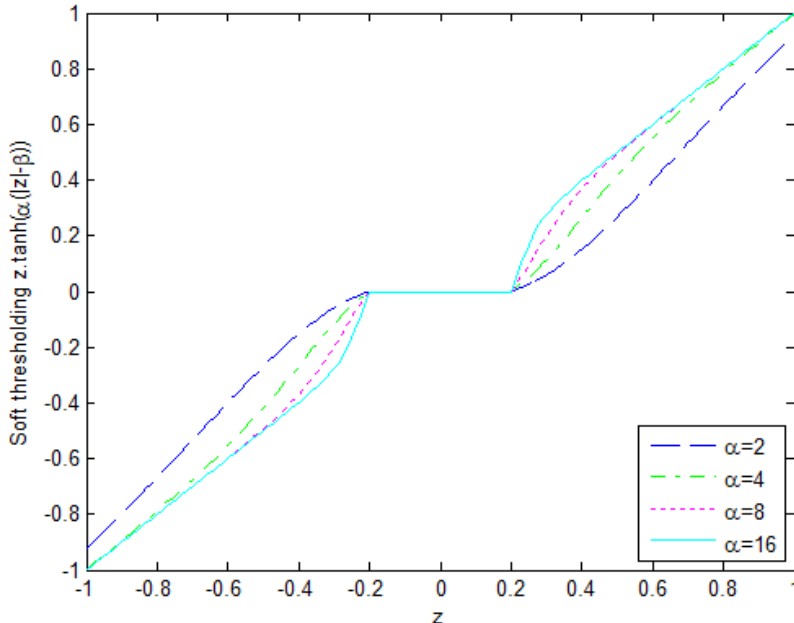

**Figure 4.** Hyperbolic tangent function based thresholding for alpha $\alpha$ = (2, 4, 8, 16), the value of $\alpha$ determines the slope of soft thresholding. The proposed method gives us the flexibility to shape the curves using $\alpha$ depending upon its application.

## 4. Results and Discussions

In order to gauge the performance of proposed algorithm, we have applied our algorithm to 1-D sparse signal, Compressively sampled MR image and Cine Cardiac MRI. MRXCAT simulator is used to evaluate the proficiency of recovery algorithms in the field of Cardiac MRI. We have evaluated the performance of our proposed technique quantitatively and qualitatively. The performance measures that are used in this research article are: pictorial depiction of under-sampling artefacts, Structural Similarity (SSIM), Peak Signal to Noise Ratio (PSNR) and Root Mean Square Error (RMSE).

### 4.1. 1-D Sparse Signal Recovery

The proposed algorithm is applied for the recovery of the 1-D sparse signal recovery, where the random sparse signal of length $n$ = 512 is created in MATLAB and the support for the sparse signal was generated randomly with K = 85 non-zero elements. The random sparse signal is compressively sampled using a random measurement matrix $A \in R^{256 \times 512}$ with only $m$ = 256 measurements.

Figure 5 shows the fitness achieved by the proposed algorithm and soft-thresholding method. The proposed method achieved faster convergence as compared to soft thresholding. Figure 6 shows sparsity effect on successful recovery achieved by the soft thresholding and proposed algorithm. The proposed algorithm performs much better even with a higher sparsity level as compared to the soft thresholding technique. Similarly, in Figure 7, the proposed method recovered the sparse signal with great accuracy, whereas the soft thresholding technique failed to accurately recover the sparse signal. The accuracy of the proposed technique was also measured against performance measures, such as SNR, MSE and correlation, as shown in Table 1. The proposed algorithm performed much better than the soft thresholding method against all these performance measures. The time comparison for proposed algorithm is 1.57 s as compared to 1.34 s by conventional soft thresholding.

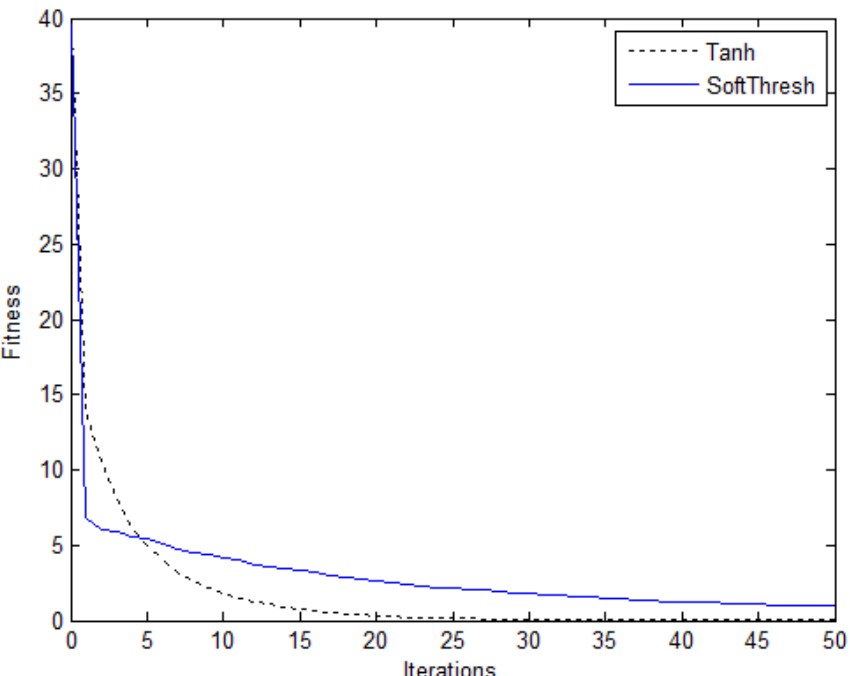

**Figure 5.** Fitness achieved by the soft thresholding and proposed algorithm. The proposed algorithm converges rapidly as compared to the soft thresholding technique.

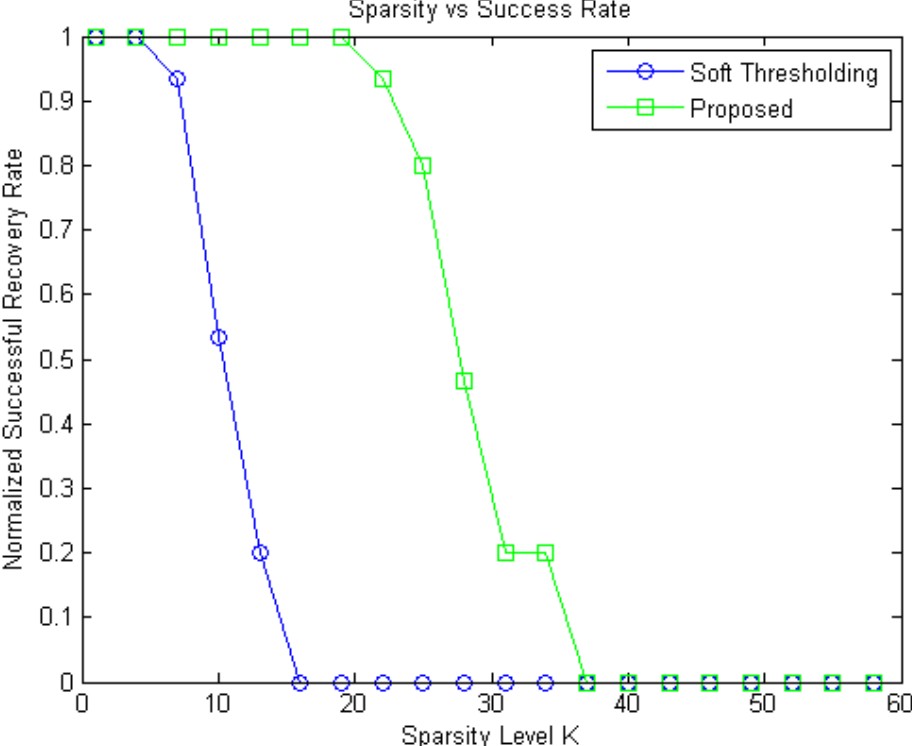

**Figure 6.** Sparsity effect on successful recovery achieved by the soft thresholding and proposed algorithm. The proposed algorithm performs much better even with higher sparsity level as compared to the soft thresholding technique.

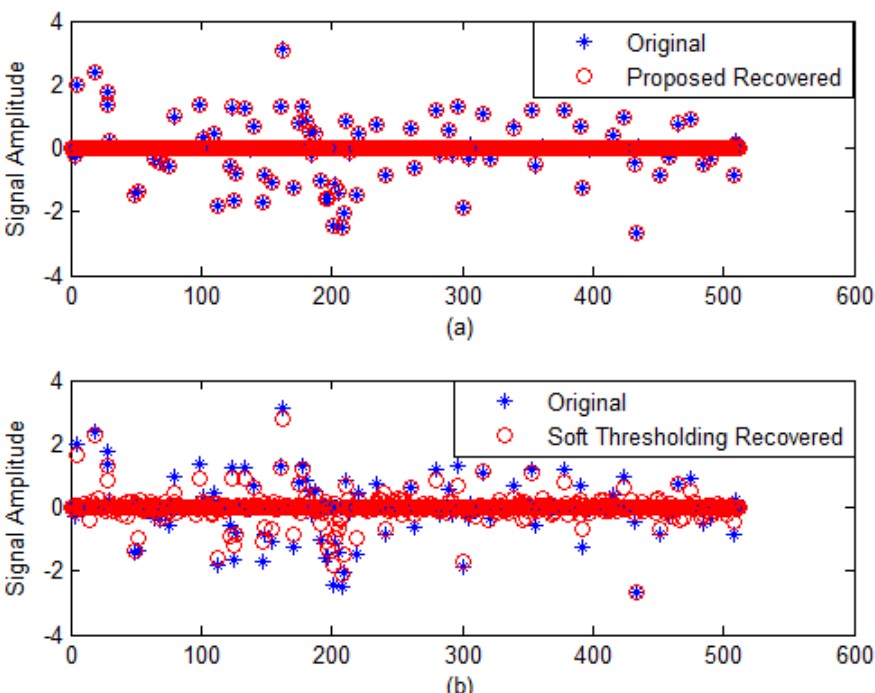

**Figure 7.** (**a**) The recovered sparse signal from the proposed algorithm; (**b**) The recovered sparse signal from soft thresholding.

**Table 1.** Performance comparison of different sparsity transforms using mean squared error in the transform domain. Temporal FFT performs better in cardiac cine MRI.

| Performance Metrics | Soft Thresholding | Proposed Algorithm |
|:---:|:---:|:---:|
| MSE | $1.00 \times 10^{-2}$ | $1.61 \times 10^{-4}$ |
| Fitness | 0.8664 | 0.0224 |
| SNR | 12.6712 | 30.6259 |
| Correlation | 0.9787 | 0.9995 |

*4.2. 2-D Compressively Sampled MR Image Recovery*

The random sampling at the CS image acquisition produces incoherent and noise-like artefacts in its sparsifying domain. In case of MR imaging or similar Fourier domain encoded biomedical imaging, where the MR image is in the spatial domain, the linear reconstruction (where, missing Fourier data points are replaced by zero and the resultant image inverse Fourier transform is taken) produces artifacts similar to additive Gaussian noise. The type of noise produced by subsampling is governed by undersampling patterns [29]. In order to recover an image, the compressed sensing recovery essentially becomes an image denoising problem. Using this analogy of CS encoding and noisy image, the first step in recovering the original image is to estimate a noise, this is achieved by maximum a posteriori (MAP) estimator. The proposed algorithm is also implemented to recover a 2-D Compressively sampled real human brain MR image of size 256 × 256. The human brain MR image is a fully sampled scanned image by a 1.5 Tesla GE-HDxt-MRI scanner with Gradient Echo (GE) sequence and 8 channels head coils with the specifications, i.e., TE = 10 msec, flip angle = 90°, bandwidth = 31.25 KHz, slice thickness=3 mm, TR = 55, and image dimensions = 256 × 256, at St. Mary's Hospital, London, UK. This MR image is compressively sampled by taking only 25% samples in k-space.

Figure 8 shows the performance of the proposed method with respect to Structural Similarity (SSIM). The proposed method achieved much better SSIM as compared to soft thresholding. The Peak Signal to Noise Ratio (PSNR) accomplished by proposed method is shown in Figure 9. Figure 10 shows the (a) Original 2D Brain MR Image,

(b) Conventional Soft Thresholding based recovered 2D MRI, (c) 2D Brain MR Image recovered from undersampled image, (d) Difference of original and soft thresholding image, (e) Difference of proposed recovery method image with original image. The Difference is scaled up by 1000 in order to enhance its visibility. Table 2 shows the proposed method has outperformed soft thresholding method in terms of PSNR and SSIM. Table 3 shows the performance of proposed algorithm and soft thresholding in terms of Mean Square Error (MSE), Improved Signal to Noise Ratio (ISNR), Correlation, SSIM, SNR, PSNR after the 15 iterations of soft thresholding and the proposed algorithm. The results show that proposed method achieved much better results in as compared to soft thresholding.

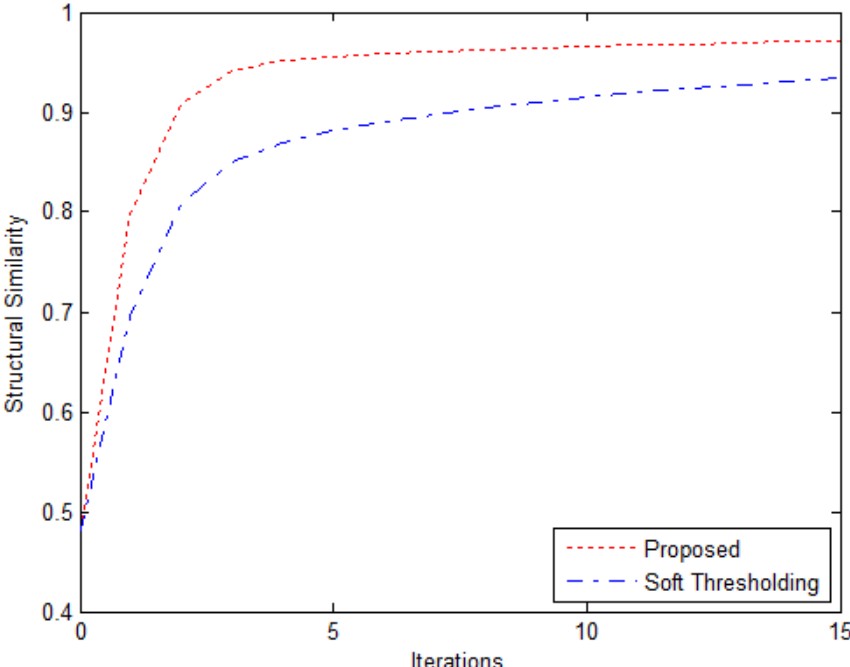

**Figure 8.** Structural Similarity of proposed and soft thresholding algorithm for recovery of compressively sampled MR image against each iteration.

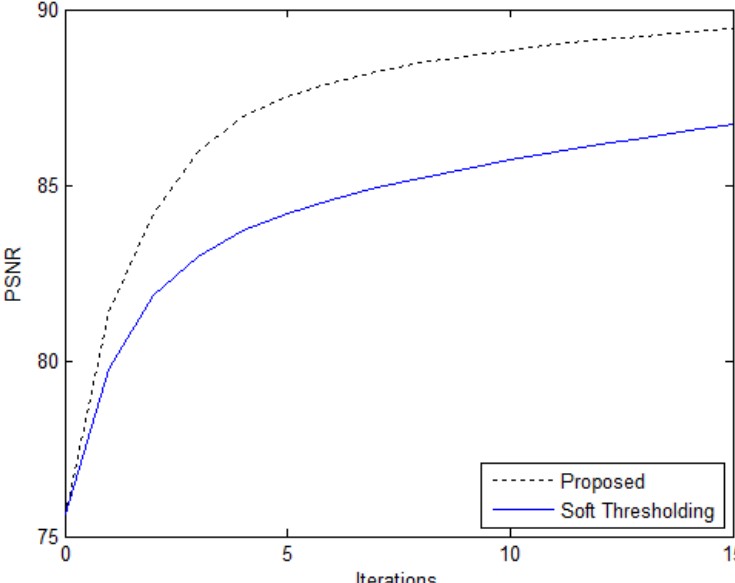

**Figure 9.** Correlation of proposed and soft thresholding algorithm of recovered compressively sample MR image.

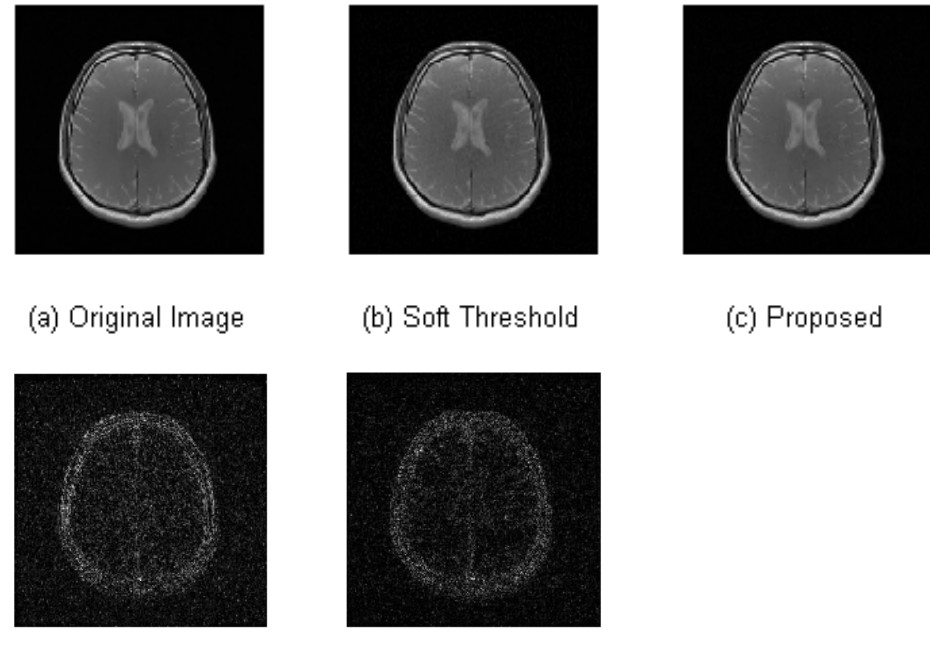

(a) Original Image  (b) Soft Threshold  (c) Proposed

(d) Difference Soft Thresholding  (e) Difference Proposed Method

**Figure 10.** (**a**) Original 2D Brain MR Image, (**b**) Conventional Soft Thresholding based recovered 2D MRI, (**c**) 2D Brain MR Image recovered from undersampled image, (**d**) Difference of original and soft thresholding image, (**e**) Difference of proposed recovery method image with original image. The Difference is scaled up by the factor of 1000 in order to enhance its visibility.

**Table 2.** Performance comparison of conventional soft thresholding and proposed method with different compression levels, i.e., 5% to 50% of subsampling of the original 2-D MR image. These results show that the proposed method achieves better results in terms of SSIM and PSNR at varying compression ratios.

| Compression Ratio | Soft Thresholding | | Proposed Algorithm | |
|---|---|---|---|---|
| | SSIM | PSNR | SSIM | PSNR |
| 5 % | 0.6843 | 75.9056 | 0.7048 | 76.1609 |
| 10% | 0.7786 | 78.9320 | 0.8175 | 79.6580 |
| 20% | 0.8994 | 82.0316 | 0.8472 | 83.7628 |
| 30% | 0.9407 | 87.3535 | 0.9790 | 91.1620 |
| 40% | 0.9724 | 91.2540 | 0.9920 | 96.1281 |
| 50% | 0.9884 | 95.4245 | 0.9955 | 99.5496 |

**Table 3.** Performance comparison of different sparsity transforms using mean squared error in the transform domain. Temporal FFT performs better in cardiac Cine MRI.

| Performance Metrics | Soft Thresholding | Proposed Algorithm |
|---|---|---|
| MSE | $1.38 \times 10^{-4}$ | $0.73 \times 10^{-4}$ |
| PSNR | 86.7195 | 89.4497 |
| ISNR | 28.3832 | 31.1135 |
| SSIM | 0.9346 | 0.9711 |
| SNR | 26.0298 | 28.7491 |
| Correlation | 0.9980 | 0.9989 |

*4.3. Cardiac Cine Magnetic Resonance Imaging Recovery*

The proposed algorithm is applied to MRXCAT, which produces breath-held undersampled cardiac cine MR image data. For MRXCAT, the following parameters were set:

recovery matrix size: 256 × 256 with 24 cardiac phases, with an image resolution set as 1 × 1 × 1 mm³, TR = 3 ms, TE = 1.5 ms. Five different acceleration rates R = (2, 4, 8, 12, 20) were used to assess the performance of the proposed method. For in vivo data, the following parameters were used: reconstruction matrix size: 256 × 256, 25 cardiac phases, with FOV of 375 mm. TE = 1 ms, TR = 3 ms and flip angle = 600. Five acceleration rates are $R = (2, 4, 8, 12, 20)$ are used to evaluate the performance of the proposed method. The reconstructed images are matched with the fully sampled original generated cardiac cine MRI as shown in Figure 11. All images are recovered in MATLAB by the proposed algorithm.

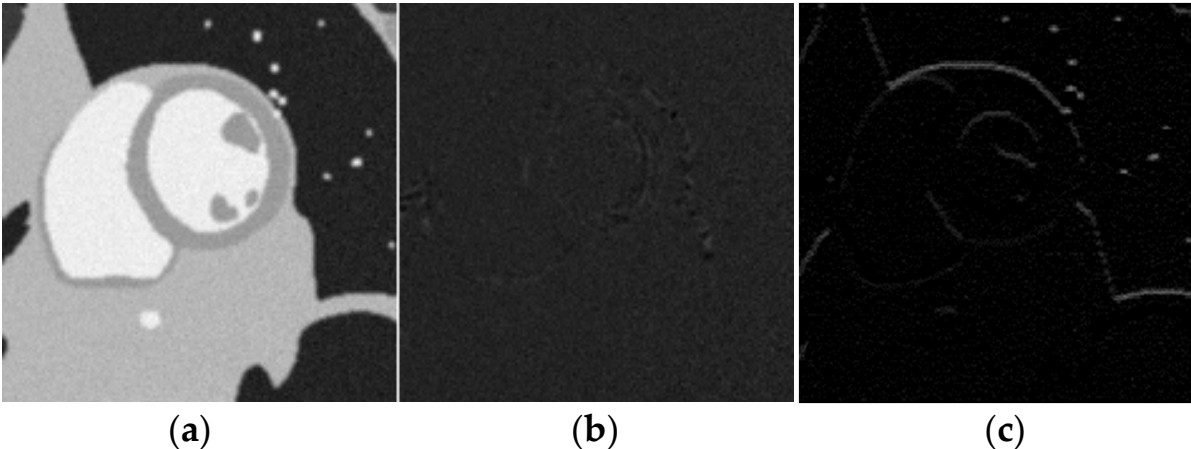

**(a)**            **(b)**            **(c)**

**Figure 11.** (**a**) Short axis cardiac cine MRI with completely sampled diastolic frame. (**b**) Sparsifying transform of cine cardiac MRI diastolic frame with temporal Fourier transform (Ψ), which results in sparse representation, (**c**) Another sparse representation of cardiac cine MR image (diastolic frame) using total variation transform (Ψ).

To gauge the efficiency of proposed algorithm, we use MRXCAT simulator software. It is designed for the analysis of reconstruction algorithms performance in the area of cardiac cine MRI. MRXCAT simulator is used to evaluate the proficiency of recovery algorithms in the field of Cardiac MRI. We have evaluated the performance of our proposed technique quantitatively and qualitatively. The performance measures that are used in this research article are: pictorial depiction of under-sampling artefacts, Structural Similarity (SSIM), Peak Signal to Noise Ratio (PSNR) and Root Mean Square Error (RMSE).

To evaluate the performance of proposed recovery technique qualitatively, we have experimentally depicted the recovered diastolic and systolic frames using acceleration rates of (R = 2, 4, 8, 12, 20). The quantitative assessment of the proposed algorithm is done using RMSE, PSNR and SSIM. Comparison between proposed algorithm and traditional soft thresholding technique is also performed. Figure 11 depicts the proficiency of the proposed algorithm at various acceleration rates while comparing with the soft thresholding. The first column shows the diastolic frame at different acceleration rates of cine cardiac MR image and the second column represents the systolic frame of cine MRI. The top row depicts the results of traditional IST algorithm, while the bottom row depicts proposed method results.

Table 4 shows the performance comparison of different sparsity transform using mean squared error in the transform domain. It measures the average of the error squares between the reconstructed and the acquired coefficients in the sparse domain. The proposed tangent hyperbolic based approximation performs well in temporal FFT as compared to the other sparse transforms. In particular, at higher acceleration rates, the tangent hyperbolic tangent based proposed technique shows much improved recovery of CS images.

**Table 4.** Performance comparison of different sparsity transforms using mean squared error in the transform domain. Temporal FFT performs better in cardiac cine MRI.

| Acceleration Rates | Spatial Domain | Total Variation | Temporal FFT |
|---|---|---|---|
| 2 | 0.1096 | 0.1123 | 0.0728 |
| 4 | 0.2321 | 0.1849 | 0.0848 |
| 8 | 0.2810 | 0.2438 | 0.0948 |
| 12 | 0.3533 | 0.2684 | 0.1043 |
| 20 | 0.4756 | 0.2982 | 0.1150 |

Figure 12 shows simulated data where (a) compares proposed method at bottom row with IST at top row with acceleration rate of 2. The arrow in (a) depicts the very minute presence of artefacts, while (b) depicts the performance of proposed method at acceleration rate equal to 4. The arrow in (b) depicts the presence of artefacts (c) shows the results of both algorithms with acceleration rate set at 8. The artefacts due to subsampling become gradually more visible in IST results as highlighted by arrow mark (d) depicts the results when acceleration rate is set at 12. Both techniques depicts the artefacts, however these artefacts are visible in the IST as mentioned by the white arrow in the figure (e) shows very much degraded image quality of IST, while comparing it with the proposed method. The subsampling artefacts dominate the traditional IST result when acceleration rate R is set at 20.

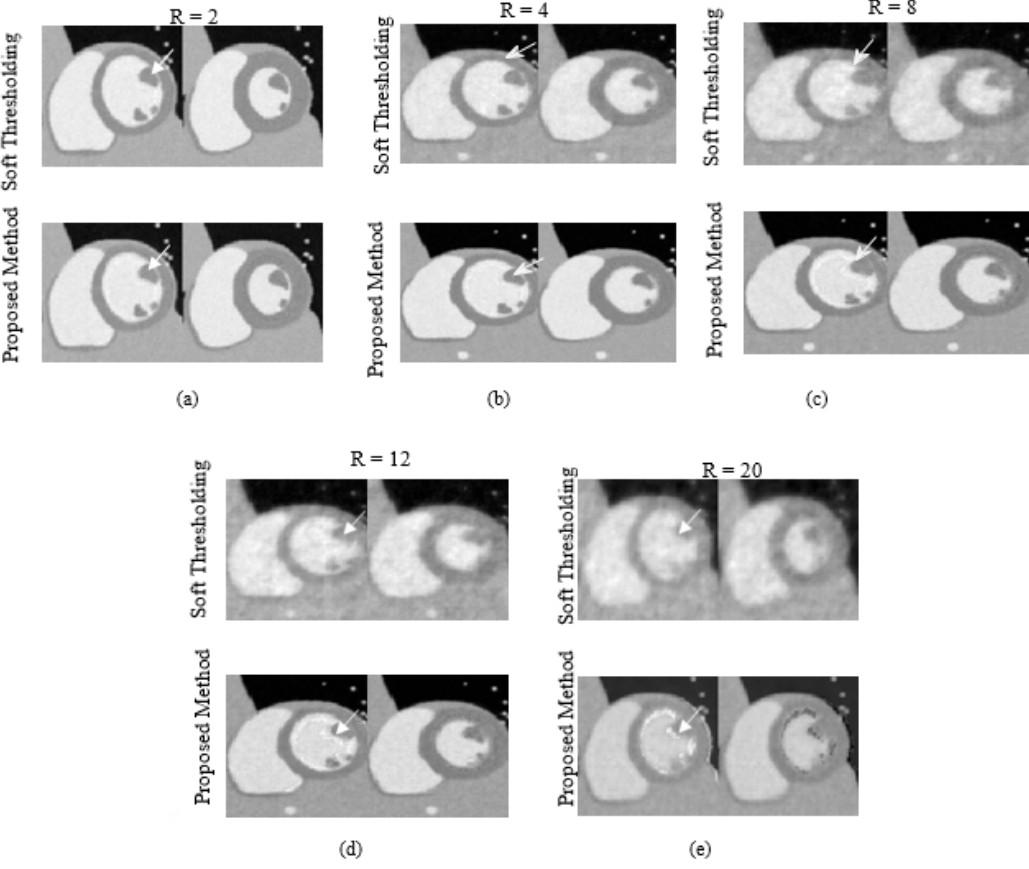

**Figure 12.** Simulated data (**a**) Compares proposed method at bottom row with IST at top row with acceleration rate of 2. The arrow in (**a**) depicts the very minute presence of artifacts. Here (**b**) depicts the performance of proposed method at acceleration rate equal to 4. The arrow in (**b**) depicts the presence of artefacts (**c**) shows the results of both algorithms with acceleration rate set at 8. (**d**) depicts the results when acceleration rate is set at 12. These artefacts are visible in the IST as mentioned by the white arrow in the figure (**e**) shows very much degraded image quality of IST, while comparing it with the proposed method.

To evaluate the recovered images quantitatively, we have used SSIM to compare the proposed technique with the IST technique. Figure 13 depicts the SSIM of our proposed algorithm, iterative soft thresholding (IST) and undersampled images. The quality of undersampled images is visibly quite poor. The efficiency of our proposed technique and IST technique is almost similar at low acceleration rates. However, the visible quality of the soft thresholding based recovered images decreases as acceleration rates are increased, as compared to the proposed method.

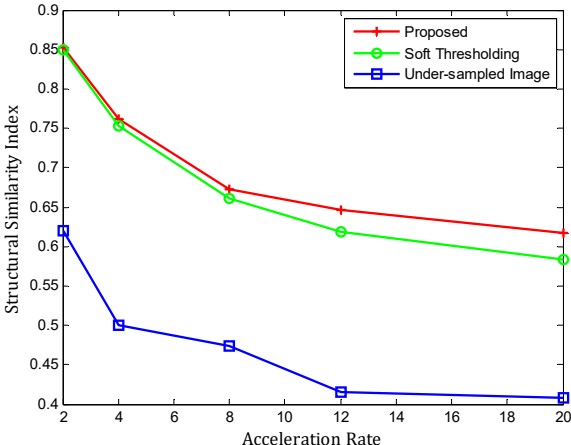

**Figure 13.** This figure depicts the efficiency of proposed algorithm by means of SSIM index. As acceleration rate increases, the SSIM of proposed algorithm degrades slowly while comparing it with IST algorithm.

Figure 14 shows the requisite iterations for the image reconstruction in both the methods. The proposed method solves the problem in six iterations, while the IST recovery technique takes ten iterations to reach the optimal solution. In this result, the data consistency is used to show the performance of proposed method.

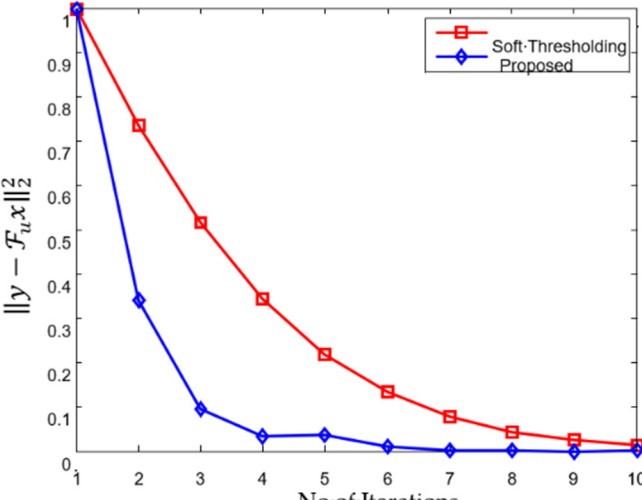

**Figure 14.** Comparison of cardiac cine MRI recovery at number of iterations; it can be seen from results that proposed method converges to an optimal solution in lesser iterations as compared to traditional thresholding.

To evaluate the efficiency of our recovered algorithm quantitatively, the results are shown the reconstruction results using PSNR at various acceleration rates (R = 2, 4, 8, 12, 20). We have compared our method with the traditional iterative soft thresholding technique. Figure 14 depicts the efficiency of our proposed technique at various acceleration rates

as compared to IST algorithm. Red line shows the results of our method at the different acceleration rates. Figure 15 shows the PSNR of soft thresholding method and under-sampled data are shown with the green and blue lines, respectively.

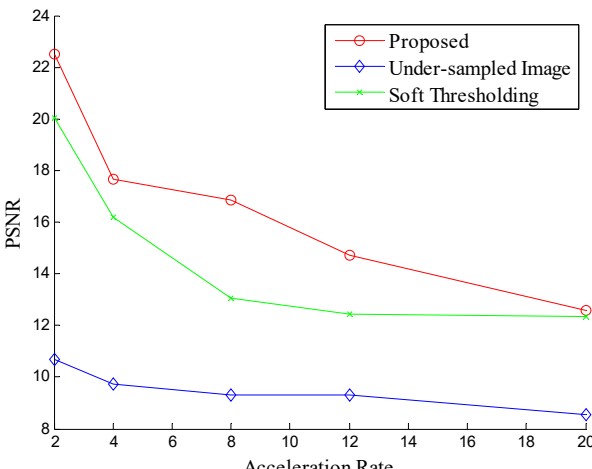

**Figure 15.** The performance using the peak signal-to-noise ratio (PSNR). PSNR of our method is better at all acceleration rates as compared to the soft thresholding method.

Table 5 elaborates the comparison of performance using root mean squared error (RMSE). It measures the error squares average between the recovered samples and the actual samples. The proposed method performance is much superior as compared to the IST technique. In particular, while operating at higher acceleration rates, the tangent hyperbolic method performs much better in recovering the images. However, the efficiency of traditional IST algorithm degrades at higher acceleration rates.

**Table 5.** Comparison of proposed method with conventional IST algorithm with RMSE. Proposed method performance is much better as acceleration rates are increased.

|  | Acceleration Rates | Undersampled Image | Iterative Soft Thresholding | Proposed Method |
|---|---|---|---|---|
| Simulated Data | 2 | 0.081 | 0.0365 | 0.0353 |
|  | 4 | 0.1218 | 0.0472 | 0.0372 |
|  | 8 | 0.1498 | 0.0702 | 0.0419 |
|  | 12 | 0.1583 | 0.0775 | 0.0485 |
|  | 20 | 0.1782 | 0.0941 | 0.0606 |
| In vivo Data | 2 | 0.085 | 0.0099 | 0.0056 |
|  | 4 | 0.106 | 0.0241 | 0.0172 |
|  | 8 | 0.1170 | 0.0495 | 0.0206 |
|  | 12 | 0.120 | 0.0567 | 0.0338 |
|  | 20 | 0.1398 | 0.0585 | 0.0551 |

To evaluate the performance of reconstructed images qualitatively, using in vivo data, we have shown a comparison between the proposed method and the soft thresholding method in Figure 16. We have used five acceleration rates R = (2, 4, 8, 12, and 20) to show the comparison between the proposed method and the soft thresholding method. In this figure, the performance of the proposed technique and IST algorithm is similar at lower acceleration rates. However, the blurring artefacts in IST based recovered images are more prominent at higher acceleration rates as compared to the proposed algorithm as indicated by the white arrows.

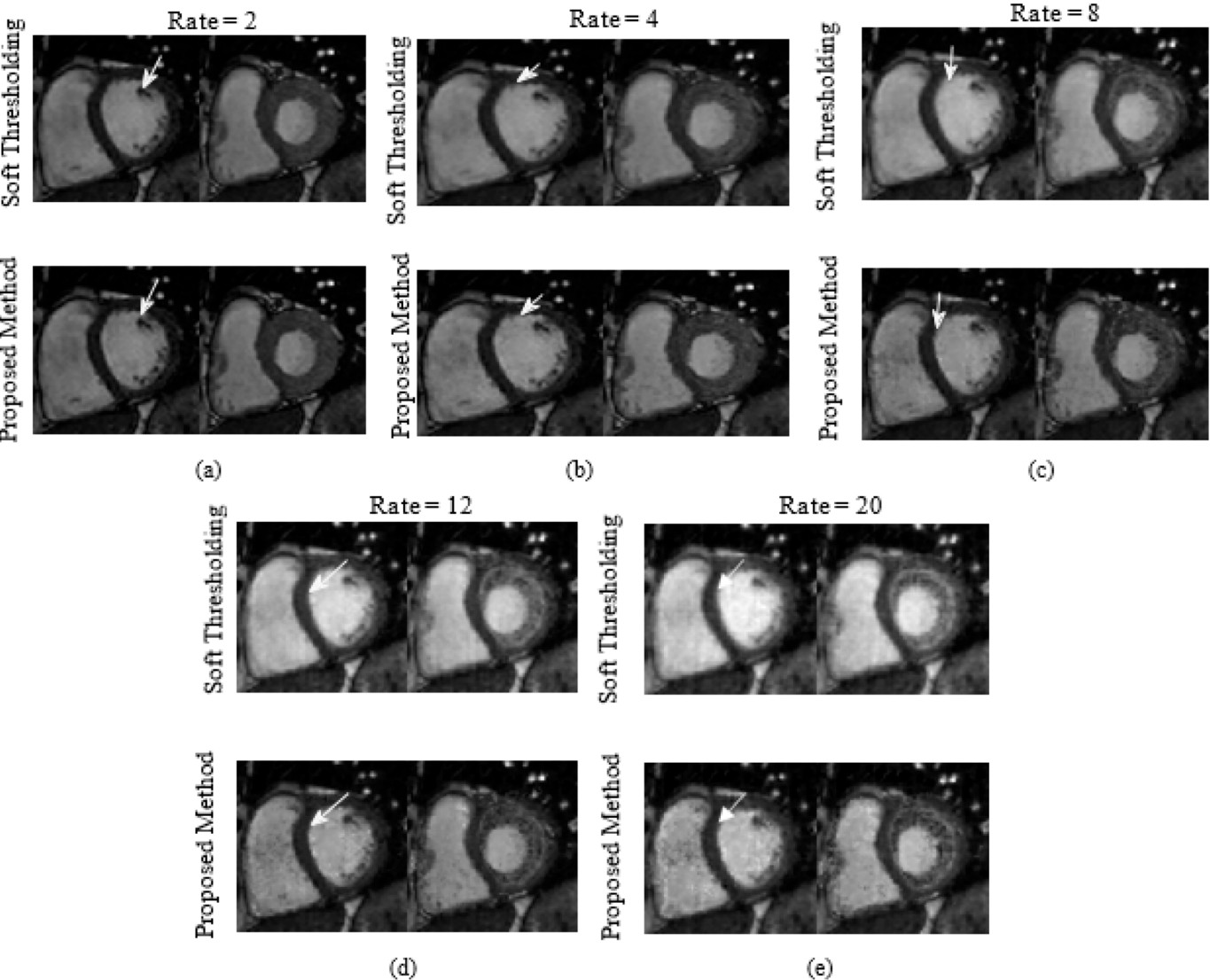

**Figure 16.** In real vivo data (**a**) Compares proposed method at bottom row with IST at top row with acceleration rate of 2. The arrow in (**a**) shows very minute artifacts. (**b**) depicts the performance of proposed method at acceleration rate equal to 4. The arrow in (**b**) depicts the presence of arte-facts (**c**) shows the results of both algorithms with acceleration rate set at 8. The artefacts due to subsampling become gradually more visible in IST results as highlighted by arrow mark (**d**) depicts the results when acceleration rate is set at 12. Both techniques depicts the artefacts, however these artefacts are visible in the IST as mentioned by the white arrow in the figure (**e**) show very much degraded image quality of IST, while comparing it with the proposed method. The subsampling artefacts dominate the traditional IST result when acceleration rate R is set at 20.

## 5. Conclusions

In this paper, the novel CS recovery algorithm is proposed for compressively sampled sparse signals and biomedical images. The proposed method is applied in the 1-D sparse signal, 2-D real human brain MRI and cardiac cine MRI. In our proposed algorithm, we have introduced a hyperbolic tangent smooth approximation of non-differentiable $l_1$-norm and shrinkage. The experimental results quantitative analysis based on SSIM, PSNR, RMSE of recovered sparse signal and MR images have outperformed the conventional IST algorithm. The qualitative observations show significant improvement in the proposed method, especially at higher acceleration rates on Cine Cardiac MR images. In future, this research work can be further enhanced to incorporate machine learning techniques using

large compressively sampled MRI datasets to restore accurate images, and the proposed method can be implemented on CS MRI scanners to reduce patient anxiety.

**Author Contributions:** Conceptualization, H.H. and J.A.S.; methodology; software, H.H.; validation, K.K., N.K. and J.A.S.; formal analysis, H.H.; investigation, H.H. and J.A.S.; resources, J.A.S.; writing—original draft preparation, H.H.; writing—review and editing, J.A.S. and N.K.; visualization, H.H. and J.A.S.; supervision, J.A.S.; project administration, J.A.S. and K.K.; funding acquisition, K.K. All authors have read and agreed to the published version of the manuscript.

**Funding:** This research was partially funded by British Malaysia Intitute, Universiti of Kaul Lumpur, including publication of this research article.

**Data Availability Statement:** 1-D dataset was generated using MATLAB. 2-D MR image was acquired from St. Mary's Hospital, London, England. While breath-held under-sampled cardiac cine dataset is generated using MRXCAT software.

**Acknowledgments:** We acknowledge the support given by University of Kaula Lumpur, British Malaysian Institute for funding the research publications. We also acknowledge Higher Education Commission Pakistan for providing support in terms of hardware for experiments.

**Conflicts of Interest:** The authors declare no conflict of interest.

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
