# Peer review of "Sparse Reconstruction Using Hyperbolic Tangent as Smooth l1-Norm Approximation"

_computation, doi:10.3390/computation11010007_

Round 1

Reviewer 1 Report

In this paper authors present an new compressed sensing reconstruction algorithm and applied it to Cardiac cine MRI images.

Abstract should contain some obtained numerical results.

MRI images in the paper are low resolution and differences are very hard to notice. Some kind of 2D difference map between the original image and CS images should be given to emphasize the differences.

Conclusion is missing future work and should give more information about the obtained results.

Author Response

I would like to thank reviewer for going through my research paper and pointing out valid suggestions for improvement of this research article. I have tried my level best to in cooperate all the points in my revised version. I would like to respond each point as following:

  1. Abstract should contain some obtained numerical results.

Response: I have modified the abstract and added discussion regarding numerical results of my experimental work.

  1. MRI images in the paper are low resolution and differences are very hard to notice. Some kind of 2D difference map between the original image and CS images should be given to emphasize the differences.

Response: I have replaced some low resolution images with better quality images in revised version. I have also added 2D MRI images differences of original image and compressively sampled recovered image in Figure 9 of research article.

  1. Conclusion is missing future work and should give more information about the obtained results.

Response: Thank you for highlighting this deficiency in research article, I have modified the conclusion section where major experimental results are summarized and some of the future directions for research fraternity are also added.

Hopefully, reviewer will be satisfied with the improvements I have made during limited time period using his suggestion and I am once again thankful for your invaluable suggestions.

Reviewer 2 Report

In this paper, authors have proposed a new CS reconstruction algorithm for compressively sampled sparse signal recovery. The proposed method is applied in the 1-D sparse signal, 2-D real human brain MRI and cardiac cine MRI. In this method, they introduced a hyperbolic tangent-based approximation of non-differentiable ?1-norm and shrinkage. The method gives better results based on qualitative and quantitative analysis, as compared to the conventional soft thresholding based reconstruction method. 

In my opinion, this paper is well organized and results are interesting. I have a following query to the authors. 

What is the advantage of considering ?1-norm instead of ?p-norm (1\leq p<\infty)? They should mention it in the Introduction part as well as in the Conclusion.

Author Response

I would like to thank reviewer for going through my research paper and pointing out valid suggestions for improvement of this research article. I have tried to in cooperate all the suggestions in my revised version of research article. I would like to respond each point as following:

  1. In this paper, authors have proposed a new CS reconstruction algorithm for compressively sampled sparse signal recovery. The proposed method is applied in the 1-D sparse signal, 2-D real human brain MRI and cardiac cine MRI. In this method, they introduced a hyperbolic tangent-based approximation of non-differentiable ?1-norm and shrinkage. The method gives better results based on qualitative and quantitative analysis, as compared to the conventional soft thresholding based reconstruction method. In my opinion, this paper is well organized and results are interesting. In my opinion, this paper is well organized and results are interesting.

Response: I would like to thank reviewer for going through research article and appreciating our research effort.

  1. What is the advantage of considering ?1-norm instead of ?p-norm (1\leq p<\infty)? They should mention it in the Introduction part as well as in the Conclusion.

Response: Thank you for your valuable suggestion to include some discussion on the different regularization norms. I have included this discussion from row number 79 to 86 of revised version of this research article.

Hopefully, reviewer will be satisfied with the improvements I have made using his suggestion and I am once again thankful for your invaluable input.

Reviewer 3 Report

This manuscript proposed an interesting approximation algorithm based on hyperbolic tangent function for the sparse reconstruction of MRI images. The hyperbolic tangent function is a particular radial basis function. The properties of this approximation (particularly the smoothness at original points and a far-away points) are investigated. Comparison with other reconstruction algorithms is made through some examples. Generally, the research in this manuscript is of interests to the readers, but its writing-up should be improved and some clarifications are necessary. My particular comments and suggestions are as follows:

1.      Sparsity for reconstruction and incoherent sampling are used in manuscript. Their quantitative description or measure should be given.

2.      For the examples, the sensitivity of the sparsity and incoherent sampling should be analyzed to demonstrate the advantages of this algorithm.

3.      Computation efficiency should be demonstrated except for its convergence. This is because the basis function is complicated for computation.

4.      English and presentations: Generally, this manuscript is understandable but presentations are confusing. Some grammatical errors can be found.

Author Response

I would like to thank reviewer for going through my research paper and pointing out valid suggestions for improvement of this research article. I have tried to in cooperate all the suggestions in my revised version of research article. I would like to respond each point as following:

  1. Sparsity for reconstruction and incoherent sampling are used in manuscript. Their quantitative description or measure should be given.

Response: Thank you for pointing out this to improve my research article I have analyzed the proposed algorithm with varying sparsity level of 1-D sparse signal. I have also added Table 3 at 334 of revised article, where sampling compression ratio is varied from 5% to 50% and proven that proposed method performs better at various compression ratios.  

  1. For the examples, the sensitivity of the sparsity and incoherent sampling should be analyzed to demonstrate the advantages of this algorithm.

Response: Indeed both sensitivity of sparsity and incoherent sampling are key aspects in CS framework. I would like to thank you for your invaluable input. Keeping in view its importance I have added effect of varying sparsity levels and incoherent sampling with varying compression ratios. It has been shown experimentally that in both cases proposed algorithm outperforms the conventional soft thresholding method.

  1. Computation efficiency should be demonstrated except for its convergence. This is because the basis function is complicated for computation.

Response: The computational efficiency of proposed technique is evident from Figure 6, where fitness achieved in only 20 iterations gives better results than 50 iterations of conventional soft thresholding method. In Figure 8, only 4 iterations are required to reach the near optimal result in terms of Structural Similarity, where conventional method fails to reach in 15 iterations. In Figure 9, PSNR achieved by proposed method in 5 iterations is much better than achieved by conventional method in 15 iterations. Computation time for both algorithms is added in row number 287 of revised article.

  1. English and presentations: Generally, this manuscript is understandable but presentations are confusing. Some grammatical errors can be found.

Response: I have improved English proficiency in consultation with English language experts and hopefully there will be no deficiency found in revised version.

Hopefully, reviewer will be satisfied with the improvements I have made during limited time period using his suggestions and invaluable input. I would once again thank the reviewers for his invaluable suggestions.
